# The Delineation and Ecological Connectivity of the Three Parallel Rivers Natural World Heritage Site

**DOI:** 10.3390/biology12010003

**Published:** 2022-12-20

**Authors:** Hui Li, Wanqi Guo, Yan Liu, Qiman Zhang, Qing Xu, Shuntao Wang, Xue Huang, Kexin Xu, Junzhi Wang, Yilin Huang, Wei Gao

**Affiliations:** 1College of Forestry and Landscape Architecture, South China Agricultural University, Guangzhou 510642, China; 2College of Architecture and Planning, Yunnan University, Kunming 650091, China

**Keywords:** ecological connectivity, minimum cost distance model, ecological function, Three Parallel Rivers Natural World Heritage Site

## Abstract

**Simple Summary:**

The scientific delineation and conservation of natural heritage sites are important steps in maintaining biodiversity and the dynamic stability of ecosystems. We examined the ecological connectivity of the Three Parallel Rivers Natural World Heritage Site, analysed the impact of boundary adjustments on the landscape pattern and ecological connectivity of the area, and explored the relationship between landscape connectivity and species habitat conservation using the Yunnan golden snub-nosed monkey as an environmental indicator species. The ecological connectivity of the Nu–Lancang River area and the Lancang–Jinsha River area was always high. Meanwhile, the improvement in ecological connectivity in the Lancang–Jinsha River area was conducive to the migration and reproduction of large terrestrial animals, including the Yunnan snub-nosed monkey, with increasing numbers of populations and individuals. In this study, we found that, in order to improve the level of conservation of heritage sites, the delineation of the effective conservation areas of protected areas and the landscape connectivity between them should be considered in addition to the total area of the protected region. In particular, in addition to large wildlife habitat protection patches, stepping stones and ecological corridor networks should be established for large terrestrial animals.

**Abstract:**

Landscape connectivity refers to the degree of continuity between the spatially structured units of a landscape. Ecological connectivity can characterise the degree to which ecological functional areas are connected in terms of function and ecological processes. In this study, the landscape pattern index and ecosystem service values were used to evaluate the ecological functional resistance of each landscape type, taking the Three Parallel Rivers Natural World Heritage Site as an example and the habitat distribution and population size of the Yunnan snub-nosed monkey as a reference. The minimum cost distance model, combined with the barrier impact index (BEI) and ecological connectivity index (ECI), was used to determine the degree of barrier impact on the study area and the ecological connectivity of the core reserve of the heritage site in both 2000 and 2020. The resistances of the different land types and landscape heterogeneity to the ecological function of species migration between the core protected areas of the heritage site were, in descending order, those of the forest, shrubs and grass, water, unused land, cultivated land, and built-up land. In 2020, the study area had a large BEI, with areas such as built-up areas, major roads, the sides of large rivers, and arable land being significant contributors to the blockage of landscape connectivity. The overall landscape connectivity in the study area was generally low, with clear spatial differentiation and a three-column parallel distribution pattern influenced by the topography and landscape. With the adjustment of the core reserve boundaries of the heritage site, the proportion of areas with high connectivity (ECI = 4–5) increased from 11.31% in 2000 to 34.36% in 2020. This increased landscape connectivity was conducive to the migration and reproduction of large terrestrial animals, such as the Yunnan snub-nosed monkey, with increasing numbers of populations and individuals. This study provides theoretical and methodological insights into the delineation and conservation of natural heritage sites and landscape connectivity.

## 1. Introduction

From an aesthetic or scientific point of view, natural heritage refers to natural landscapes consisting of physical and biological structures or groups of such structures that have outstanding universal value. From a scientific or conservation point of view, natural heritage refers to geological and physiographic structures of outstanding universal value and clearly defined areas of threatened animal and plant habitats. From a scientific, conservation, or natural beauty point of view, natural heritage refers to natural attractions of outstanding universal value or clearly delineated natural areas [1]. Therefore, the scientific delineation and conservation of natural heritage sites are important steps in maintaining biodiversity and the dynamic stability of ecosystems. The use of landscape map partitioning and clustering methods, as well as the combination of graphical metrics with detailed geographic information and the behavioural characteristics of organisms in the landscapes, can reveal the most suitable target species habitats, migration paths, and key landscape elements for constructing the corridors, patches, and background landscape components of ecosystem composition [2,3,4]. Common delineation methods based on this theory include the minimum cost distance model zoning method and landscape resistance surface analysis [5,6,7]. The conservation areas of natural heritage sites delineated on this basis are often large, complex, and unstable spatiotemporal structures [8]. In addition, there are multiple interests and stakeholders in the management of protected areas [9], leading to serious conflicts in human–land relations; these external threats and internal conflicts have profound impacts on the ecological integrity of natural heritage sites. The conservation of ecosystem integrity has become the main basis for boundary delineation, which emphasises whether key ecological features, such as the composition, structure, and function of ecosystems, can be restored and developed healthily within a certain regional context in the face of various types of disturbances [10,11]. Landscape connectivity can be used to analyse the ecosystem integrity of natural heritage site boundaries by examining the organic linkages in functions and ecological processes between similar or dissimilar patches in the landscape [12]. In recent years, an increasing number of scholars have used landscape and landscape connectivity assessments at the landscape scale to study and explore nature reserves [13,14,15,16].

The concept of landscape connectivity was first used by Merriam in 1984 to describe the interaction between the structural features of a landscape and the movement behaviour of species. Landscape structure refers to the variety, diversity, quantitative composition, and spatial and hierarchical relationships of landscape components and the characteristics of their influencing factors [17]. Forman and Godron (1986) considered landscape connectivity as a measure that describes how corridors or substrates in a landscape are connected and continued spatially [18]. Taylor (1993) considered landscape connectivity to be the extent to which the landscape facilitates or hinders the movement of organisms or some ecological process between source–sink patches [19]. With et al. (1997) described landscape connectivity as the functional relationship between habitat patches caused by the spatial spread of patches and the movement of organisms in response to landscape structure [20]. Marulli and Mallarach (2003) proposed a new model for evaluating landscapes and landscape connectivity at a regional scale using the ecological connectivity index and the barrier effects index to diagnose connectivity in terrestrial landscape ecosystems. This model not only allows for a cost–benefit assessment of the current situation, but also has predictive capability to quantitatively assess and compare the impacts of different planning scenarios or different infrastructure alternatives on the landscape and its connectivity [21]. At present, research on landscape connectivity has rapidly developed in terms of theory, methods, and applications, mainly focusing on quantitative evaluation methods for landscape connectivity and the interaction between connectivity and ecological processes, as well as the exploration of different models [22,23,24]. In terms of applied research, the main focus has been on the migration and dispersal of organisms, ecological fragmentation, and the construction of ecological safety patterns [25,26,27], while ecological processes such as material and energy flow and disturbance propagation have been given less attention.

Due to the integrated consideration of dynamic processes and ecological processes in landscapes, landscape connectivity research methods have attracted considerable attention and discussion in the field of landscape ecology. These methods are generally divided into three categories. Experimental research methods utilize data on the movement or distribution of organisms to determine the most direct and realistic degree of functional connectivity. Such methods focus on different species of organisms and use biotelemetry, mark-release retrieval, and mass marker retrieval to follow the movement paths of individuals in detail, thus obtaining direct data on the behavioural characteristics of the organisms [13,28,29,30,31,32,33,34]. The second category includes quantitative approaches to research using landscape pattern indices, which are simple quantitative indicators that reflect certain aspects of the structural composition and spatial configuration of a landscape by condensing information about its pattern at a high density. The landscape pattern index is an index used in ecological studies to measure the spatial structural characteristics of a landscape [35]. This category of methods is divided into two main types: landscape pattern indices and models that are used either directly as connectivity indices or transformed into indices [21,36,37,38]. The third category includes model simulations of specific ecological processes, which allow for the further use of limited data to abstract and simplify real systems; these methods occupy an important place in the study of landscape ecology. In the study of landscape connectivity, the types of models used mainly include landscape dynamics models, composite population models, and migration–diffusion models [39,40,41,42,43]. Among them, the landscape pattern index method can reflect the apparent spatial continuity of a certain landscape type by knotting a structural connectivity index that includes parameters such as fragmentation, agglomeration, spreading, separation, and connectivity. Combined with ecological connectivity and connectivity probability measures based on distance threshold settings, and connectivity models based on minimum cumulative resistance, this method can also reveal the extent to which a landscape type facilitates or hinders various ecological processes [44,45,46,47]. It is not only able to describe landscape connectivity quantitatively, but can also identify areas and components with significant landscape connectivity. For larger study areas, it can reflect the structure of a patch in relation to the nature of the patch; however, ecological process thresholds and other elements of the composite index are more applicable in practical applications.

The Yunnan snub-nosed monkey (scientific name: *Rhinopithecus bieti*), a species of monkey from the genus *Rhinopithecus*, is one of the 25 rare and endangered primates in the world; it is listed as an endangered species on the IUCN Red List of Threatened Species, is a Class I protected animal under the Washington Convention, and is a part of the Red Book of Endangered Animals of China. The monkey mainly inhabits a small area in the Yunling Mountains of China between the Lancang and Jinsha rivers in the Three Rivers Region [48], and almost all extant populations of Yunnan snub-nosed monkeys are isolated and island-like [49]. The monkey mainly lives in primitive fir forests at high altitudes (above 3000 m); however, their activities can range from 2500 m to 5000 m in high mountains, low scrub, meadows, and flowing rocky beaches. Lichen pine loosestrife growing on fir trees acts as a stable back-up food in their diet, which also includes the young leaves of conifers, overwintered flower buds and leaf buds, plant shoots, and young leaves [50]. Yunnan snub-nosed monkeys are an iconic endemic large mammal species in this region, and their population dispersal and genetic diversity have become difficult to maintain due to their specific food requirements and habitat sensitivity. The integrity of the snub-nosed monkey’s habitat and the connectivity between habitats can be regarded as a sign of the soundness of the ecosystem in the Three Parallel Rivers region. Since its approval in 2003, the Three Parallel Rivers Natural World Heritage Site has undergone a number of boundary adjustments. It is uncertain whether these adjustments have affected the integrity and landscape connectivity of the site. It is important to determine whether boundary adjustments have had corresponding impacts on the conservation and migration of large species, such as snub-nosed monkeys, in this area. The level of landscape connectivity is an important factor in maintaining the integrity, sustainability, and stability of natural ecosystems. Therefore, the ecological connectivity method was used in this work to explore the relationship between landscape connectivity and large species habitat conservation in the region, using the Yunnan snub-nosed monkey (the most sensitive species) as an environmental indicator. First, we used the landscape index method to analyse the spatial and temporal landscape differentiation, landscape pattern characteristics, and ecosystem service function values of the Three Parallel Rivers Natural World Heritage Site. Secondly, we used the minimum depletion distance model to analyse the barrier impact index and the level of ecological connectivity within the site. The aim of this study was to explore the following issues: (1) the characteristics of landscape pattern changes before and after boundary adjustments; (2) the impact of boundary adjustments on landscape connectivity; and (3) the impact of changes in landscape connectivity on the populations of large species, such as the Yunnan snub-nosed monkey.

## 2. Materials and Methods

### 2.1. Study Area Overview

The Three Parallel Rivers Natural World Heritage Site, officially approved as a Natural World Heritage Site at the 27th World Heritage Conference on 2 July 2003, is a site in the northern hemisphere where all types of natural landscapes—except for desert and marine landscapes—are present, meeting all four of the Natural World Heritage Site assessment criteria [51]. It refers to the area where the Nu, Lancang, and Jinsha rivers flow parallel to each other in western Yunnan (Figure 1). It contains a wide range of climate and biome types, equivalent to the southern subtropical, central subtropical, northern subtropical, warm temperate, temperate, cold temperate, and cold zones of the northern hemisphere, and is a microcosm of the bio-ecological environment of the Eurasian continent. Furthermore, it is the region with the most dramatic differentiation of biological species and communities since the early Cenozoic Era and one of the most biodiverse regions in the world, ranking first among the 17 “key regions” for biodiversity conservation in China. It is home to the vast majority of biome types in the northern hemisphere and has one of the largest numbers of ecosystem types of any region in the world [43]. The 2001 Natural World Heritage Site is an important and ecologically valuable conservation area selected by the IUCN after a study commissioned by United Nations Educational, Scientific, and Cultural Organization (UNESCO). The site covers a total area of about 17,000 km^2^, of which the core area covers 9394 km^2^ and consists of ten scenic areas and five nature reserves (Figure 3a).

However, the report from a joint World Heritage Committee and IUCN expedition in 2006 showed that the region was experiencing ecological problems: increased human–land conflicts due to mining, tourism, hydropower development, and boundary modifications; threats of narrow ecosystem zones, habitat fragmentation, and poor ecological stability; and glacier degradation and snow line rise as a result of global climate change. These issues threaten the authenticity and integrity of the heritage site, and the region is at risk of being added to the List of World Heritage in Danger. The boundaries of the Three Parallel Rivers Natural World Heritage Site have been subjected to several rounds of adjustments, and the boundaries of much of the site have been significantly altered in connection with economic activities such as mineral development and hydropower development. The Red Hill area has seen the most drastic adjustment, giving way to 16 large- and medium-sized mining areas, including the Pulang copper polymetallic mine. This adjustment reduced the core of the Red Hill area by approximately 222 km^2^, reducing the total area from 1648 km^2^ to 1426 km^2^ (a reduction of 13%), whereas the buffer zone was reduced from 2608 km^2^ to 1446 km^2^. In August 2010, at the 34th session of the World Heritage Committee, the boundary revision report submitted by China in 2009 was narrowly approved by one vote. The total area of the core area of the Three Parallel Rivers Natural World Heritage Site increased from 9394 km^2^ in 2003 to 9600 km^2^, and the total area of the buffer zone increased from 7584 km^2^ to 8164 km^2^ (Figure 3b).

### 2.2. Research Ideas

The research ideas and process used in this study are shown in Figure 2.

### 2.3. Data Sources

In this study, 30 m land use data for 2000 and 2020 from GlobeLand30 (http://www.globallandcover.com/ (accessed on 5 March 2022)) were used to classify the study area into six categories of landscape types: forest, shrubs and grass, water, cultivated land, built-up land, and unused land (snow and bare ground) (Figure 3). The overall accuracy of the data was higher than 85%, and the results of a comparison with SkyMap satellite images showed that the data were consistent with the actual situation.

### 2.4. Research Methods

The ecological functions of each landscape type were refined into landscape pattern indices, which were also assessed in combination with the values of the ecosystem service functions, and the entropy value method was used to determine the index weights [52]. The entropy method is a mathematical method used to determine the degree of dispersion of a certain indicator. The greater the degree of dispersion, the greater the influence of that indicator on the comprehensive evaluation [53] for superimposition to determine the ecological function resistance. The minimum cost distance model was then used to combine the barrier impact index (BEI) and ecological connectivity index (ECI) to calculate the degree of barrier impact in the study area for both 2000 and 2020, and the ecological connectivity between the landscape areas and nature reserves was measured based on the natural heritage site boundaries in 2001 and 2011 (Table 1). Finally, the results were visually represented using GIS (geographic information science) technology [54,55].

### 2.5. Analysis of Land Use Characteristics and Ecological Functions in the Study Area in 2000 and 2020

Based on the Yunnan snub-nosed monkey’s habitat and migration characteristics, the land use characteristics and ecological function analysis of the study area from 2000 to 2020 are summarized in Table 2 below.

Among the landscapes in the study area, non-forested lands, barren mountains, and Yunnan pine forests are unsuitable areas for Yunnan snub-nosed monkeys, while snow-covered lands, agricultural and grazing lands, artificial construction sites, planted economic forests, and especially large rivers—such as the Nu, Lancang, and Jinsha rivers, which lie between the mountains—are the main obstacles to the migration of the monkeys [48,62]. Therefore, Yunnan snub-nosed monkeys are mainly distributed in the area between the Lancang and Jinsha rivers and are unable to migrate across these rivers into the vast areas to the west of the Lancang River or east of the Jinsha River (Figure 4).

### 2.6. Analysis of Resistance to Ecological Functions in the Study Area in 2000 and 2020

First, each landscape index was comprehensively considered to characterise ecological functions in the study area, such as landscape connectivity, disturbance dispersal, restoration, and ecological vulnerability. The area of the patch type (CA) (Equation (1)), number of patches (NP) (Equation (2)), sub-dimensionality (PAFRAC) (Equation (3)), and landscape condensation index (COHESION) (Equation (4)) were then selected as the landscape structure influencing factors of ecological function intensity and combined with the calculation of the ecological service function values of the landscape types (Equation (5)) (Table 1).

Secondly, the entropy value method (Equations (6)–(9)) was used to comprehensively evaluate the influences of the structure and ecosystem service functions of different landscape types on the flow of ecological functions in the study area, resulting in the weights for the patch type area (CA), number of patches (NP), sub-dimensionality (PAFRAC), landscape condensation (COHESION), and ecosystem service function value (*V*) being determined as *a* = 0.2768, *b* = 0.0909, *c* = 0.1271, *d* = 0.1327, and *e* = 0.3725, respectively.

Finally, the landscape pattern index and ecosystem service value data were combined based on the entropy value method weighting calculation, according to Equation (10). The ecological function intensities of the study area in order from strongest to weakest were forest > shrubs and grasses > water > unused land > cultivated land > built-up land. In addition, in an ecological function network, the ecological function resistance of each landscape type is inversely proportional to its ecological function intensity (Equation (11)), which can be derived from the ecological function resistance of each landscape type, as shown in Table 3.

### 2.7. Barrier Impact Index and Ecological Connectivity

Based on the characteristics of the effects of different types of land use on the habitat and life processes of the snub-nosed monkeys in Yunnan and the magnitude of the ecological function resistance calculated above (Table 3), we ultimately selected the top five land use types in the resistance ranking as the barrier land use types in the Three Parallel Rivers Natural World Heritage Site and assigned weighting values based on relevant papers [21,63] (Table 4).

Using the five barrier types mentioned above as sources, the parameters of the resistance level of the site types calculated by the minimum cumulative resistance model (Table 4) were determined according to the ranking of ecological functional resistance (Table 5), in consultation with relevant experts and based on the habits of the snub-nosed monkeys. In this regard, because unused land, such as water and permanent and non-permanent snowy land, is the biggest obstacle to the movement of large terrestrial animals, including the snub-nosed monkey, the level of obstruction was set to be equal to that of built-up land. The barrier effect *Ys* for each of the five barrier land types was calculated according to Equation (12). Then, the total barrier effect (BEI) was calculated using Equation (13). Finally, the BEI was equally spaced into five levels to represent the barrier impact index: level 1 indicates very low impact, level 2 denotes low impact, level 3 represents medium impact, level 4 indicates high impact, and level 5 is very high impact.

After the calculation of the barrier impact index was completed, the ecological connectivity index (ECI) was calculated based on the minimum cumulative resistance model using the boundaries of the natural heritage site from 2001 and 2011 as ecological sources and the BEI classification results from 2000 and 2020 as resistance surfaces. The resulting ecological connectivity values were divided into five classes in equal intervals from small to large.

## 3. Results

### 3.1. Analysis of the Impact of Barriers in the Study Area

As shown in Table 6, against the background of a high area ratio of natural vegetation in the study area (90%), only 3.33% and 6.56% of the region had high barrier impact levels (BEI = 4–5) in 2000 and 2020, respectively. However, there were more areas with medium to high barrier impact levels in 2020 than in 2000, indicating that the overall landscape ecology of the study area has been affected by increased levels of barrier disturbance as human activities, such as urban construction and arable land development, have intensified.

In terms of spatial distribution (Figure 5), the areas with higher barrier impacts were mainly located in the Three Parallel Rivers watersheds, built-up areas, and on both sides of the main roads in the study area. Medium impact areas included transition areas between the roads and forest and cultivated areas. Some of the areas with a larger vegetation concentration had larger proportions of surrounding built-up land, and several of these larger areas were also threatened by a greater level of obstacle disturbance because of surrounding construction land.

The results of the analysis of the structure of the landscape components in the different barrier-affected areas are shown in Figure 6. The proportions of cultivated land, water, and built-up land increased significantly with the degree of impact, whereas the areas of forest, shrubland, and grassland decreased with the degree of barrier impact; thus, natural vegetation, as the dominant component of the study area’s landscape composition, was the main factor that determined the spatial grading of impact areas at all levels. The proportion of forests in the low and medium BEI areas was about 70%, whereas in the very high BEI areas it decreased to around 55%. In contrast, built-up land, the most heavily disturbed component of the study area, was concentrated in the high and very high BEI areas; the data from both 2000 and 2020 showed that over 80% of built-up land was in high and very high impact areas. Cultivated land was distributed in both medium and very high impact areas, with the percentages gradually increasing from 7.54% and 7.18%, respectively, in 2000 to 12.61% and 14.00% in 2020 with increasing levels of barrier impact.

### 3.2. Analysis of Landscape Connectivity

Based on the statistical results of the proportion of ecologically connected areas in the study region (Table 7), the overall ecological connectivity in the study area has increased; the proportion of highly connected and very highly connected areas in the study area was 11.31% in 2000, while the proportion of areas with high connectivity or more increased to 34.36% in 2020. According to the spatial distribution of the ECI (Figure 7), the data from the two years showed that areas with high connectivity were mainly located east of the Nu River and west of the Jinsha River, whereas areas with low connectivity were mainly located in the eastern part of the study area in the scenic area of Qianhu Mountain, the area around the scenic area of Haba Snow Mountain, the southern side of Meili Snow Mountain in the northwest, and the northwestern part of Gaoligong Mountain Nature Reserve. Compared with that in 2000, the ecological connectivity of the northern part of the study area in 2020 has increased because of the combination of Gongshan Scenic Area, Meili Snow Mountain Scenic Area, Julong Lake Scenic Area, and Baima Snow Mountain Nature Reserve; in addition, north–south ecological connectivity has increased due to the addition of Yunling Nature Reserve in the south. However, the Hong Mountain Scenic Area and Qianhu Mountain Scenic Area in the east have reduced in scope and changed in form. Consequently, in the surrounding areas, ecological connectivity with other ecological function areas has been reduced because of the reduction in the scope of protection and morphological changes.

Because of the specific topography and landscape of the Three Parallel Rivers Natural World Heritage Site, the biggest migration barriers for large terrestrial animals, represented here by Yunnan snub-nosed monkeys, are the large rivers, including the Nu, Lancang, and Jinsha rivers. Yunnan snub-nosed monkey populations are only distributed within a narrow strip of land between the Lancang and Jinsha rivers; the monkeys are able to migrate north–south in this region but are unable to cross the Jinsha and Lancang rivers to the east and west. The increase or decrease in ecological connectivity across the study area cannot be explained by its effect on a particular terrestrial population. Therefore, the study area was divided into four regions: the west bank of the Nu River, the Nu–Lancang River area (the area between the Nu and Lancang rivers), the Lancang–Jinsha River area (the area between the Lancang and Jinsha rivers), and the east bank of the Jinsha River (Table 8).

As shown in Table 8, the connectivity of the area on the west bank of the Nu River is low because the river barrier has made it more difficult to communicate and connect with the area east of the Nu River. The north–south connectivity of the area west of the Nu River has increased because of the conservation of the Gaoligong Mountain Nature Reserve, the Yueliang Mountain Scenic Area, and the Pianma Scenic Area, with the proportion of medium connectivity areas increasing by 8.12%. By 2020, a significant increase in north–south landscape connectivity was observed between the Nu–Lancang River area and the Lancang–Jinsha River area, which basically covers the habitat of all Yunnan snub-nosed monkey populations between the Jinsha and Lancang rivers. In particular, the habitat of the Yunnan snub-nosed monkey population in the Yunling area has improved relative to the habitat of other populations in the central part of the area. The ecological environment of the various populations has improved, leading to an increase in habitat quality. Landscape connectivity in the eastern region of the Jinsha River remains at a low to medium level, and the area of very low connectivity has expanded to the north as the relative distance between the shrinking eastern ecological function area and other ecological function areas increases.

## 4. Discussion and Strategy

The results of the study show that quantitative analysis of ecological connectivity demonstrated that the study area has low landscape connectivity and significant spatial differentiation, and because of the local topography, there is a three-column parallel distribution pattern of ecological connectivity. The east–west ecological connectivity is consistently low, whereas the north–south ecological connectivity has increased, with the proportion of high connectivity areas (ECI = 4–5) rising from 11.31% in 2000 to 34.36% in 2020. By comparing and analysing the changes in ecological connectivity in the four regions, we determined the factors influencing the migration dispersal of terrestrial large mammals, such as snub-nosed monkeys, as shown in Table 9. The ecological connectivity of the habitats of the various populations of snub-nosed monkeys in the Lancang–Jinsha River region improved, which contributed to the improvement of the habitat quality level. The establishment and adjustment of nature reserves have provided good conditions for the survival and reproduction of Yunnan snub-nosed monkeys. From 1996 to 2020, the population size of snub-nosed monkeys in Yunnan increased from 13 to 23, and the number of individuals steadily increased from 1500 to 3845 [64,65,66].

The results showed that human beings have effectively intervened in the maintenance of ecological environment and the restoration of connectivity through the establishment of nature reserves and the adjustment of boundaries, promoted interspecific communication and gene flow opportunities of indicator species, and improved the survival and diffusion ability of species [67]. Meanwhile, our analysis of the barrier impact index showed that because forest land, shrubland, and grassland make up over 90% of the overall landscape in the study area, there was already a degree of landscape fragmentation; however, it was at a low level and the landscape ecology was less disturbed. However, with urban expansion, as well as the expansion of agricultural land and related infrastructure projects, the level of barrier impact in 2020 increased compared to that in 2000. The areas with greater barrier impacts are now distributed in a typical network shape, mainly in the centres of built-up areas, on both sides of major roads and large rivers, and in areas where arable land is concentrated, which is an important reason for the blockage of ecological connectivity. Therefore, the rational planning of the road transport network, controlling the spatial expansion of towns and villages, and the avoidance of oversized and contiguous patches of agricultural production will be conducive to the improvement of ecological connectivity.

Based on the results of the study, further analysis showed that the current distribution of Yunnan snub-nosed monkey populations in the Lancang–Jinsha River region still exists over a long distance between the Yunling Nature Reserve, Laojun Mountain Scenic Area, and Baima Snow Mountain Nature Reserve. However, snub-nosed monkeys in Yunnan are socially biotic animals. The characteristics of the spatial structure of survival are closely related to feeding, the avoidance of natural predators, and access to mating in snub-nosed monkey communities [68]. The temporal and spatial distribution of food determines the size of individual population habitats, which ranges from 14.1 to 18 km^2^ in spring, 9.5 to 18 km^2^ in summer, 9.3 to 12.1 km^2^ in autumn, and 12.3 km^2^ in winter [69,70]. The births of infant monkeys are mostly concentrated in March–April, with a gestation period of about 7 months, where females have one litter every two years [71]. Due to differences in preference for vegetation types, there is no food competition with animals such as macaques *(Macaca mulatta)* [72]. Mammals such as clouded leopards, snow leopards, wolves, jackals, black bears, and brown bears and common buzzards, golden eagles, bearded vultures, alpine hawk eagles, kites, and owls are potential predators of Yunnan snub-nosed monkeys [73]. Yunnan snub-nosed monkeys are most sensitive to human disturbance; in particular, human deforestation activities directly affect their spatial distribution and habitat quality. In order to strengthen the landscape connectivity of the heritage site area [25], policymakers should further consider maintaining the continuity of ecological processes and patterns in the dispersal of Yunnan snub-nosed monkey populations and delineate a series of subalpine coniferous forest habitat patches with an area of 9–18 km^2^ in the direction of their dispersal in nature reserves as stepping stones and corridors for the migration of the monkeys [69,70]. The east bank of the Jinsha River has been fragmented by large-scale urbanisation and the sharp turn of the river that occurs there, which has further fragmented the already fragmented patches. However, because of the low barrier resistance around the Hong Mountain Scenic Area, as well as the fact that the Bitahai Nature Reserve is connected to the Haba Snow Mountain Scenic Area and the Haba Snow Mountain Nature Reserve to form the core reserve of the complete heritage site, the areas with low and medium ecological connectivity have been increased to a certain extent, enhancing the ecological connectivity of the area, although the overall ecological connectivity is still low. The scenic area of Qianhu Mountain and the scenic area of Haba Snow Mountain are located across the Tiger Leaping Gorge, and the ecological connectivity between the scenic area of Qianhu Mountain and the scenic area of Hong Mountain and other areas is further reduced. There is a risk of fragmentation with the surrounding environment, and the ecological connectivity of this area did not improve significantly between 2000 and 2020. Therefore, the demarcation of urban construction land expansion boundaries and ecological control lines should be accelerated, spatial allocation should be rationalised, all illegal and disorderly land development activities should be eliminated, and new development activities should be prevented from having significant negative impacts on the ecological structure of the region [36]. At the same time, necessary conservation, restoration, and rehabilitation efforts should be carried out to improve the ecological structure and enhance the landscape connectivity of the region.

In previous studies on ecological management, greater emphasis has been placed on the size of nature reserves and the proportional relationships between them. However, in this study, it was found that to improve the level of conservation of heritage sites, the delineation of the effective conservation areas of protected areas and the landscape connectivity between them should be considered in addition to the total area of the protected region. In particular, in addition to large wildlife habitat protection patches, stepping stones and ecological corridor networks should be established for large terrestrial animals such as the Yunnan snub-nosed monkey [74].

In addition to the influence of town construction, roads, and agricultural production on landscape connectivity in heritage sites [30,36], geographical gradients determine the relative importance of the environmental drivers of regional vegetation patterns and landscape connectivity [75]. The area divided by the Nu, Lancang, and Jinsha rivers is a geospatially insurmountable barrier for large mammals such as the Yunnan snub-nosed monkey. The regional conservation model of ten core scenic areas and five nature reserves, which have been designated because of urbanisation and administrative constraints, has greatly threatened the integrity and authenticity of the heritage site, exacerbated the process of landscape fragmentation, and blocked the ecological links between the various scenic areas, resulting in changes to the original ecological structure of the heritage site and hindering the migration and dispersal of species, gene flow, and other ecological processes, ultimately threatening the ecological safety and conservation of biodiversity [76]. It is worth exploring how the system of national parks could be further adjusted and improved in the future.

## 5. Conclusions

The Three Parallel Rivers Natural World Heritage Site is characterised by a complex system composed of different landscape elements or ecosystems that are organically linked through ecological processes; the overall performance of its ecosystem services is dependent on the flow of material, energy, and information between ecosystems. In this study, we use Yunnan snub-nosed monkey as an environmental indicator species. The importance of landscape connectivity for species conservation was discussed from a macro-ecological perspective based on the characteristics of species habit and spatial structure. Based on the functional integrity and connectivity of their landscape systems, the enhancement of ecosystem service functions and the value of heritage sites can be characterised to some extent by the level of conservation demarcation and landscape connectivity of those heritage sites, which, in turn, plays a role in species conservation. This study was based on the calculation of landscape pattern indices for land use types and the determination of the values of ecosystem service functions, which clarified the magnitude of ecological functional resistance; in addition, the minimum depletion distance model, combined with the barrier impact index (BEI) and ecological connectivity (ECI) research methods, was also used. This allowed for the detailed diagnosis of the connectivity of terrestrial landscape ecosystems and for the quantitative assessment and comparison of the effects of different planning scenarios or different heritage site boundary delineation scenarios on landscape and ecological connectivity. The results of this study showed the following:

First of all, landscape heterogeneity varies in its degree of resistance to species migration between the core protected areas of heritage sites, including nature reserves and scenic areas [77]. The spatial intensity of landscape function is influenced by landscape type, spatial structure, and relative distance; therefore, the typological composition of each landscape unit, its spatial configuration, and its ecosystem function directly determine its resistance to the ecological functions of patches. Moreover, the results of our comprehensive evaluation of the ecological function intensity of the study area, carried out by integrating the types, patterns, and ecological function values of the landscape, showed that the order of ecological function resistance in the study area in descending order is as follows: forest land, shrubland and grassland, water, unused land, cultivated land, and built-up land.

Secondly, because of the influence of the three large north–south rivers—the Nu, Lancang, and Jinsha rivers, which are part of the Three Parallel Rivers Natural World Heritage Site—the watershed is the biggest migration obstacle for large mammals such as the Yunnan snub-nosed monkey. Secondly, because of its alpine location, unused land, including snow, ice, and glaciers, is a major barrier to the migration of Yunnan snub-nosed monkey populations. In contrast, cultivated, road, built-up land in towns and villages and deforestation act as artificial disturbance factors for migration and reproduction of Yunnan snub-nosed monkey population [52,62]. The presence and expansion of artificial patches and corridors also puts individual ecological functional areas at risk of being isolated from each other.

Finally, by comparing the protection pathways of the four districts implemented by the government, this study showed that the scientific delineation of nature reserve boundaries and the development of corresponding conservation measures, such as rational planning of road traffic networks, control of the disorderly expansion of urban and rural spaces, reduction in human disturbances such as deforestation, and avoidance of oversized and contiguous patches of agricultural production, can provide an effective way for the conservation of large terrestrial species such as the Yunnan snub-nosed monkey.

## Figures and Tables

**Figure 1 biology-12-00003-f001:**
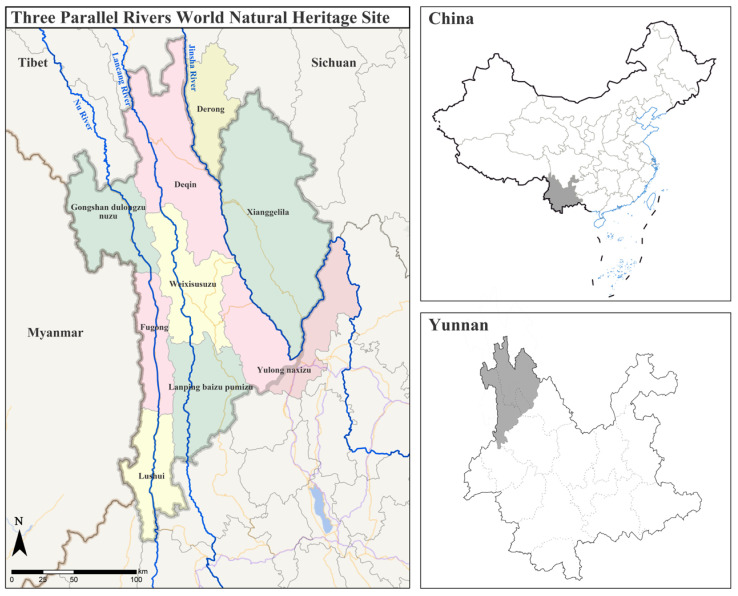
Map of the Three Parallel Rivers Natural World Heritage Site in Yunnan Province, China.

**Figure 2 biology-12-00003-f002:**
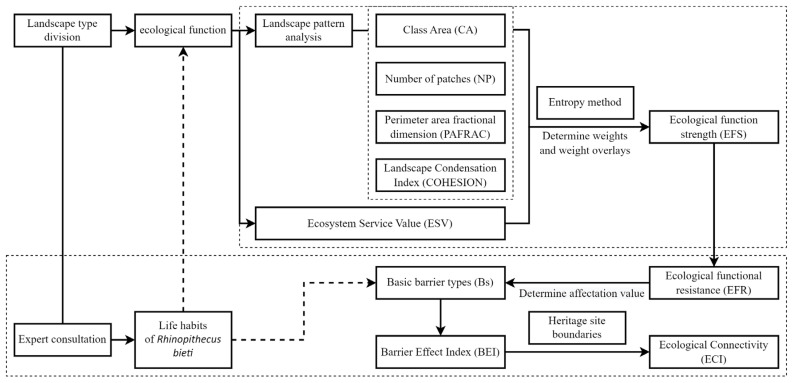
Article flowchart.

**Figure 3 biology-12-00003-f003:**
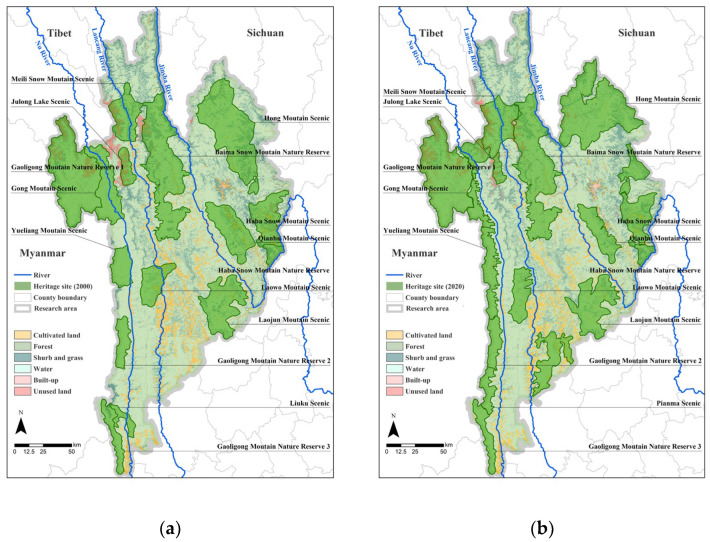
Map of the Three Parallel Rivers Natural World Heritage Site in 2000 (**a**) and 2020 (**b**). The boundaries of the Three Parallel Rivers Natural World Heritage Site are drawn with reference to the 2001 Master Plan of the Three Parallel Rivers Scenic Area of the Nu, Lancang, and Jinsha rivers prepared by the Yunnan Provincial Institute of Urban and Rural Planning and Design and the 2011 Revision of the Master Plan of the Three Parallel Rivers Scenic Area of Yunnan Province, as well as the UNESCO map of heritage sites (https://whc.unesco.org/ (accessed on 1 April 2022)).

**Figure 4 biology-12-00003-f004:**
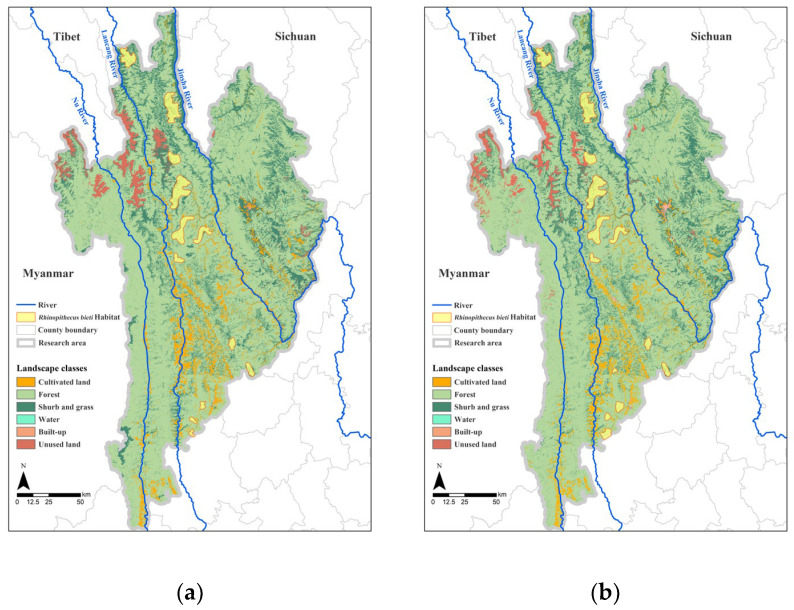
Landscape classes of the Three Parallel Rivers Natural World Heritage Site in 2000 (**a**) and 2020 (**b**). The boundaries of *Rhinopithecus bieti* habitat in 2000 and 2020 are drawn with reference to the population map of the Yunnan snub-nosed monkey released by The Nature Conservancy (https://www.tnc.org.cn/ (accessed on 10 May 2022)).

**Figure 5 biology-12-00003-f005:**
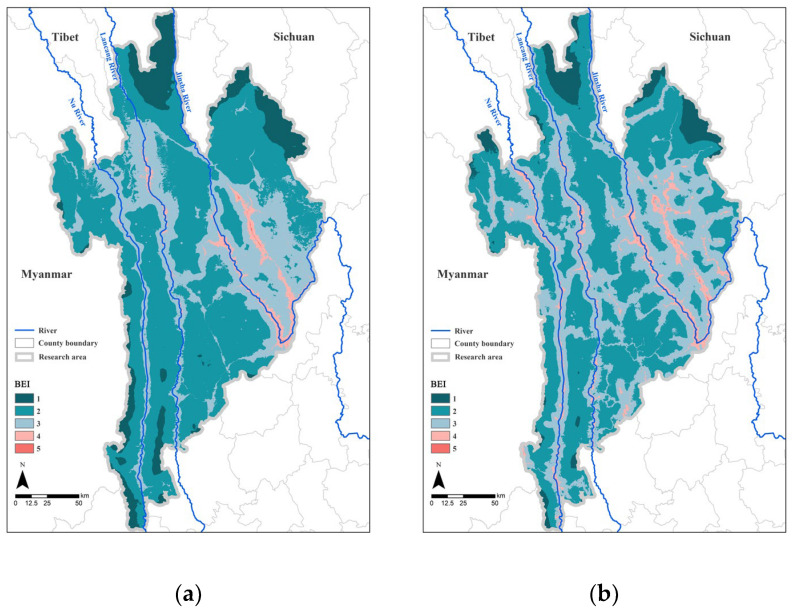
Map of the Three Parallel Rivers Natural World Heritage Site in 2000 (**a**) and 2020 (**b**) with BEI.

**Figure 6 biology-12-00003-f006:**
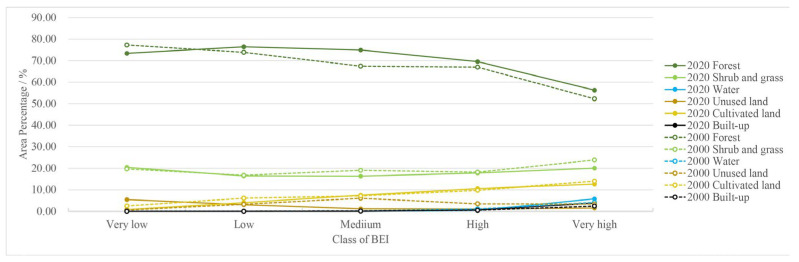
Landscape structure of different BEI areas.

**Figure 7 biology-12-00003-f007:**
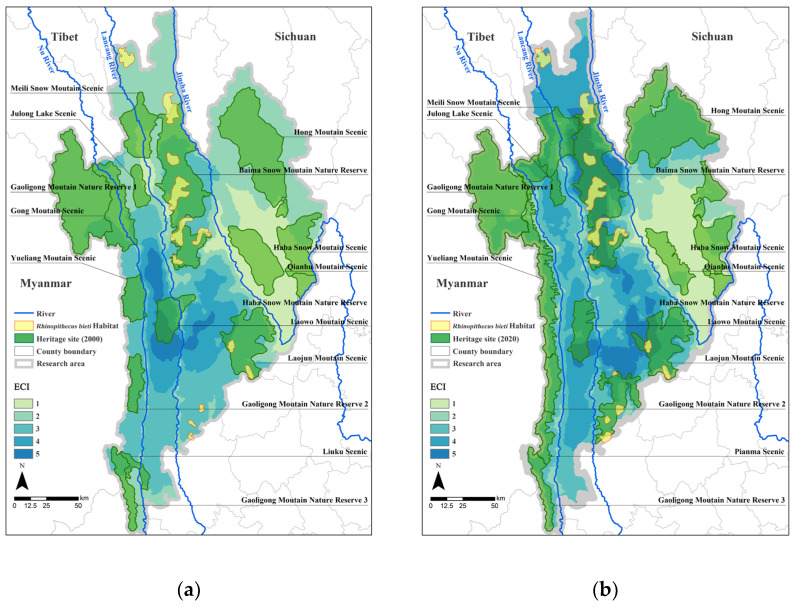
Map of the Three Parallel Rivers Natural World Heritage Site in 2000 (**a**) and 2020 (**b**) with ECI. The boundaries of the Three Parallel Rivers Natural World Heritage Site are drawn with reference to the 2001 Master Plan of the Three Parallel Rivers Scenic Area of the Nu, Lancang, and Jinsha rivers prepared by the Yunnan Provincial Institute of Urban and Rural Planning and Design and the 2011 Revision of the Master Plan of the Three Parallel Rivers Scenic Area of Yunnan Province, as well as the UNESCO map of heritage sites (https://whc.unesco.org/(accessed on 1 April 2022)). The boundaries of *Rhinopithecus bieti* habitat in 2000 and 2020 are drawn with reference to the population map of the Yunnan snub-nosed monkey released by The Nature Conservancy (https://www.tnc.org.cn/ (accessed on 10 May 2022)).

**Table 1 biology-12-00003-t001:** List of research methods, calculation results, and standardised values for the Three Parallel Rivers Natural World Heritage Site.

No.	Indicator	Indicator Definition	Calculation Method
1	Class area (*CA*)	The size of the CA value governs the abundance of species, their numbers, food chains, and the reproduction of their secondary species; the size of the area of different classes can reflect differences in the flow of energy and nutrients between species. In general, the total amount of energy and mineral nutrients in a patch is proportional to its area, affecting the degree of movement of species between resource patches [35].	(1) $$CA=\sum_{j=1}^{n}a_{ij}\times \frac{1}{100}$$ In the equation, *a_ij_* is the area of patch *ij* and takes the value range *CA* > 0, while *CA* is the sum of the areas of all the patches in a patch type, which is the total area of the patch type.
2	Number of patches (*NP*)	At the class level, NP is equal to the total number of patches of a given patch type in a landscape; at the landscape level, it is equal to the total number of all patches in the landscape. NP reflects the spatial pattern of the landscape and is often used to describe the heterogeneity of the landscape as a whole (landscape heterogeneity refers to the variability of landscape element types, combinations, and attributes in a landscape system in space or time), and the magnitude of its value is well and positively correlated with the fragmentation of the landscape [35].	(2) NP=N In the equation, *N* is the number of patches.
3	Perimeter area fractional dimension (*PAFRAC*)	Overall patch structure characteristic is used to quantify the degree of complexity and distortion of a certain landscape type. According to the ecological significance of the number of sub-dimensions, the more complex the shape of the patch, the less disturbed it is. PAFRAC can be used to characterise the ecological vulnerability of a landscape to a certain extent [35].	(3) $$FRAC=\frac{2\ln\left(0.25p_{ij}\right)}{\ln\left(a_{ij}\right)}$$ In the equation, *p_ij_* is the perimeter of the patch and *a_ij_* is the area of patch *ij*.
4	Landscape condensation index (*COHESION*)	Index measuring the spatial connectivity of a landscape type, reflecting the state of aggregation and the dispersion of patches in the landscape, with higher values indicating higher spatial connectivity [56].	(4) $$COHESION=\left[1-\frac{\sum_{j=1}^{m}p_{ij}}{\sum_{j=1}^{m}p_{ij}\sqrt{a_{ij}}}\right]{\left[1-\frac{1}{\sqrt{A}}\right]}^{-1}\times 100$$ In the equation, *a_ij_* is the area of the *j* th patch in the landscape of category *i* (m^2^); *p_ij_* is the perimeter of the *j* th patch in the landscape of category *i* (m); and *A* is the total area of the landscape (hm^2^)
5	Ecosystem service value (*ESV*)	The method chosen for calculating the service values of the different types of ecosystems in the study area was based on the unit area value equivalent factor method [57], with reference to Xie et al.’s [58,59] method for assessing the value of ecosystem services and their development of a unit area equivalent table for terrestrial ecosystems in China.	(5) $$ESV_{ij}=\sum_{j=1}^{m}A_{ij}\times VC_{j}$$ In the equation, *ESV_ij_* is the ecological service value (in RMB) of ecosystem type *j* in study area *i*; *A_ij_* is the footprint of ecosystem type *j* in study area *i*; and *VC_j_* is the ecosystem service value per unit area of ecosystem type *j* [60].
6	Entropy method	The entropy method was used to estimate the weight of each indicator, which was essentially calculated using the value coefficient of that indicator’s information; the higher its value coefficient, the greater its importance to the evaluation (or the greater the weight, the greater its contribution to the evaluation results).	(1) The data were normalized. The indicators were changed from absolute to relative values and the effect of the scale on the results was eliminated. (6) $$x_{ij}^{\prime }=\frac{x_{ij}-\min(x_{i})}{\max(x_{i})-\min(x_{i})}$$ (2) The index information entropy $E_{j}$ was calculated as follows: (7) $$E_{j}=-\frac{1}{\ln m}\sum_{i=1}^{m}p_{ij}\ln p_{ij}(i=1,2,\ldots m;j=1,2,\ldots n)$$ where $p_{ij}=\frac{y_{ij}}{\sum_{j=1}^{m}y_{ij}}$ represents the standardization indicator proportion of year i under indicator j. If $p_{ij}$ = 0, $E_{j}$ = 0. (8) $$W_{j}=\frac{E_{j}}{\sum_{j=1}^{m}E_{j}},j=1,2\cdots m$$ (3) The weight of the *j*th indicator was calculated as: where *E_j_* is the coefficient of variation of the *j*th indicator. (4) The weighted summation formula was used to calculate the evaluation value of the sample as follows: (9) $$U_{i}=\sum_{j=1}^{m}W_{j}*P_{ij}(i=1,2,\cdots n)$$ where *U* is the combined evaluation value, *n* is the number of indicators, and *W_j_* is the weight of the jth indicator. The final comparison of all *U* values led to an evaluation conclusion [53].
7	Ecological function strength (*EFS*)	By combining the values of ecosystem service functions (V) for the different landscape types, the entropy method was used to comprehensively evaluate the influence of the structure and ecosystem service functions of different landscape types on the flow of ecological functions in the study area [60,61].	(10) EFS=a×CA+b×NP+c×PAFRAC+d×COHESION+e×ESV In the equation, *CA* is the total area of the patch, *NP* is the number of patches, *PAFRAC* is the sub-dimensionality index, *COHESION* is the landscape condensation index, *ESV* is the ecosystem service value, and *a*-*e* are the weights of *CA*, *NP*, *PAFRAC*, *COHESION,* and *ESV*, respectively, calculated based on entropy values.
8	Ecological functional resistance (*EFR*)	In ecological functional networks, the ecological functional resistance of each landscape type is inversely proportional to its ecological functional intensity; the higher the ecological functional intensity, the lower the ecological functional resistance [60].	(11) $$EFR=\frac{1}{EFS}$$ In the equation, *EFS* is the ecological functional strength.
9	Barrier effect index (*BEI*)	BEI is used to express the extent to which different types of built-up land have a barrier effect on the realisation of structural or functional links between ecological land patches.	(12) $${{Ys_{i}=b}_{s}-ks}_{1}({\ln ks}_{2}(b_{s}-d_{s})+1)$$ (13) $$BEI_{i}=\sum_{i=1}^{n}Ys_{i}$$ In the equation, *Ys_i_* is the barrier effect produced by the *s*th obstacle type; *b_s_* is the weight assigned to the *s*th obstacle type; *ks_1_* and *ks_2_* are parameters used to adjust the shape of the logarithmic function curve; *ds_i_* is the depletion distance from the *i*th quadrant to the obstacle in *S* (the minimum depletion distance calculated by the minimum depletion distance model); *BEI_i_* is the barrier effect index for the *i*th quadrant; and *n* is the number of obstacle types [21].
10	Ecological connectivity index (*ECI*)	Ecological connectivity can be used to evaluate the organic linkage of ecological structures, functions, and processes between ecological functional areas. Based on the least resistance model, it considers the role of landscape matrix and corridors in ecological processes, while indirectly reflecting changes in landscape dynamics, and offers a better measure of landscape connectivity [44].	Using the map algebra function of GIS, combined with the following equations: (14) $$d_{i}=\sum_{r=1}^{m}dr_{i}\;ECI_{i}=10-9\frac{\ln\left(1+\left({d_{i}-d}_{\min}\right)\right)}{\ln{\left(1+\left({d_{\max}-d}_{\min}\right)\right)}^{3}}$$ the results of the barrier effects derived in the previous section were used as a resistance surface to calculate the landscape connectivity index for the study area. In the equation, *dr_i_* is the depletion distance from the *i*th image to the *r*th ecological type zone; *d_i_* is the total depletion distance from the *i*th image to each ecological function zone; *d_max_* is the maximum value of the total depletion distance from a given area image to each ecological function zone; *d_min_* is the minimum value of the total depletion distance from a given area image to each ecological function zone; and *ECI_i_* is the landscape connectivity index of the *i*th image [21].

Note: Landscape heterogeneity refers to the spatial or temporal variability of landscape element types, combinations, and attributes in a landscape system; landscape pattern refers to the spatial pattern of the landscape, which is the spatial distribution and combination of landscape spatial units (patches) of different sizes, shapes, and attributes; the landscape element type with the largest area and best connectivity in the landscape is defined as the landscape matrix in landscape ecology [35].

**Table 2 biology-12-00003-t002:** Analysis of land use characteristics and ecological functions in the study area from 2000 to 2020.

Type of Land Use	Percentage in 2000	Percentage in 2020	Land Use Spatial Distribution Characteristics	Ecological Function
Forest	73.20%	73.20%	The woodland is continuously distributed over a large area, and its fragmentation is the smallest among all landscape types, with the most concentrated patchy distribution in the river valley and patchy distribution in the middle- and high-elevation zones.	The woodland meets the requirements as a landscape substrate, a suitable ecological environment for the Yunnan snub-nosed monkey, and a migration corridor.
Shrubs and grass	18.95%	17.75%	Irrigation grasses are patchily distributed and most concentrated in river valleys, and patchily distributed in middle- and high-elevation zones.	The grassland is a suboptimal habitat for Yunnan snub-nosed monkeys.
Cultivated land	5.04%	5.51%	Agricultural land is distributed on both sides of rivers and roads and around cities and villages in continuous blocks, with the most regular and simple spatial shape, but with a high degree of fragmentation and greater human interference.	Agricultural land is a migratory barrier area for Yunnan snub-nosed monkeys.
Unused land	3.90%	2.80%	Unused land is concentrated in large blocks, mainly in the northwestern part of the study area, and the rocky, bare land is scattered in various types of plots in the study area.	Unused land is a migratory barrier area for Yunnan snub-nosed monkeys.
Built-up	0.14%	0.33%	Construction land is distributed in scattered patches at different elevations with a high degree of fragmentation, which shows that the residents in the study area are scattered.	Construction land is a migratory barrier area for Yunnan snub-nosed monkeys.
Water	0.28%	0.44%	The watershed is a very characteristic landscape in the study area, with the Nujiang, Lancang, and Jinsha rivers running through the whole region from north to south.	Water is a migratory barrier for the Yunnan snub-nosed monkey.

**Table 3 biology-12-00003-t003:** Landscape pattern indices, ecological function intensities, and resistance values in 2000 and 2020.

Land Cover	CA (km^2^)		NP		PAFRAC		COHESION		V (Million rmb·Hm-2)		Ecological Function Intensity		Ecological Function Resistance Value		Rank of Resistance Value
	2000	2020	2000	2020	2000	2020	2000	2020	2000	2020	2000	2020	2000	2020	
Built-up	55.88	135.71	490	764	1.30	1.28	94.09	97.24		-	-		-	-	1
Cultivated land	2063.15	2256.46	3479	3789	1.39	1.35	98.63	98.56	1898	2076	0.17	0.17	6.03	6.01	2
Unused land	1593.93	1145.71	33,050	21,617	1.39	1.40	98.77	99.03	-	-	0.20	0.22	5.01	4.48	3
Water	115.01	180.52	1056	889	1.57	1.49	98.45	99.02	9774	15,341	0.27	0.27	3.74	3.66	4
Shrubs and grasses	7097.18	7259.28	249,437	241,156	1.47	1.46	98.64	98.60	16,465	16,842	0.35	0.38	2.83	2.60	5
Forest	29,975.37	29,922.80	80,661	77,293	1.41	1.41	99.99	99.98	415,758	415,029	0.86	0.96	1.16	1.04	6

**Table 4 biology-12-00003-t004:** Basic barrier types (Bs) in the Three Parallel Rivers Natural World Heritage Site.

Code	Type	Weight (bs)	Ks1	Ks2
B1	Unused land	40	22.21	0.123
B2	Cultivated land	50	27.75	0.102
B3	Traffic road	80	44.42	0.063
B4	Urban area	100	55.52	0.051
B5	Water	100	55.52	0.051

**Table 5 biology-12-00003-t005:** Impact matrix (MA) for the calculation of BEI.

Code	Type	Affectation Coefficient	Affectation Value
V1	Forest	1000 m	0.1
V2	Shrubland and grassland	500 m	0.2
V3	Cultivated land	125 m	0.8
V4	Unused land	1 m	100
V5	Built-up land	1 m	100
V6	Water	1 m	100

**Table 6 biology-12-00003-t006:** Results of BEI classification for the Three Parallel Rivers Natural World Heritage Site.

Class of BEI	Impact	Area/km^2^		Percentage/%	
		2000	2020	2000	2020
1	Very low	3990.65	2156.34	9.76	5.27
2	Low	24,342.26	20,748.24	59.52	50.73
3	Medium	11,206.56	15,311.80	27.40	37.44
4	High	1316.88	2598.28	3.22	6.35
5	Very high	44.15	85.82	0.11	0.21

**Table 7 biology-12-00003-t007:** Results of ECI classification for the Three Parallel Rivers Natural World Heritage Site.

Class of ECI	Impact	Area/km^2^		Percentage/%	
		2000	2020	2000	2020
1	Very low	5768.25	4826.95	14.10	11.80
2	Low	17,544.69	7784.35	42.90	19.03
3	Medium	12,963.46	14,237.05	31.70	34.81
4	High	3332.76	9780.09	8.15	23.91
5	Very high	1291.37	4272.08	3.16	10.45

**Table 8 biology-12-00003-t008:** Results of ECI classification in the four parts of the Three Parallel Rivers Natural World Heritage Site.

Class of ECI	West Bank of the Nu River/%		Nu–Lancang River Area/%		Lancang–Jinsha River Area/%		East Bank of the Jinsha River/%	
	2000	2020	2000	2020	2000	2020	2000	2020
1	0.09	0.48	3.84	0.00	5.17	0.92	40.94	40.75
2	53.72	45.20	29.12	4.78	33.63	5.36	58.41	32.03
3	46.20	54.32	33.64	40.23	43.58	29.42	0.65	27.22
4	0.00	0.00	18.91	51.19	15.69	38.20	0.00	0.00
5	0.00	0.00	14.49	3.80	1.93	26.11	0.00	0.00

**Table 9 biology-12-00003-t009:** Reasons for the improvement of ecological connectivity in the subregion and related conclusions.

Regional Zoning	Comparison of Medium and High Eco-Connectivity		Ecological Connectivity Changes	Reason for Improvement	Conclusion
	2000	2020			
West bank of the Nu River	46.2%	54.32%	↑	In the 2013 master plan adjustment, Gaoligong Mountain Nature Reserve, Gong Mountain Scenic Area, Yueliang Mountain Scenic Area, and Pianma Scenic Area were linked to become the core protected area of the Natural World Heritage Site.	The consecutive protection of core reserves can effectively prevent the geographical isolation of large mammal species, providing access to feeding, courtship, and competition.
Nu–Lancang River area	33.4%	54.9%	↑	In 2012, the overall planning of the Three Rivers Scenic Area was carried out, and the northern part of the region was connected between Meili Snow Mountain Scenic Area, Gong Mountain Scenic Area, and Julong Lake Scenic Area.	Improved overall regional connectivity can provide habitats for large mammals, facilitating the migration and conservation of species.
Lancang–Jinsha River area	17.62%	64.31%	↑	(1) The Yunling Nature Reserve was established in 2003. (2) The master plan adjustment in 2013 added the Yunling Nature Reserve to the core reserve of the Natural World Heritage Site together with the eastern side of Meili Snow Mountain, Baima Snow Mountain Nature Reserve, and Laojun Mountain Scenic Area.	Provide habitat for large species such as the Yunnan snub-nosed monkey, which is conducive to the migration and protection of the species; the population size and the number of individuals are both increasing.
East bank of the Jinsha River	65%	27.22%	↓	The area of ecological protection land has been further reduced and tends to be fragmented and isolated.	Increase in migration barrier areas.

Note: ↑—eco-connectivity increase; ↓—eco-connectivity decrease.

## Data Availability

Not applicable.

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
