# Peer review of "The Delineation and Ecological Connectivity of the Three Parallel Rivers Natural World Heritage Site"

_biology, 2022, doi:10.3390/biology12010003_

Round 1

Reviewer 1 Report

This paper analyzed the changes in landscape spatial pattern between the past and present and tried to evaluate the ecological connectivity of the landscape for the population of a specific monkey. I understand this kind of study is important and the authors can analyze complex geometric patterns of the landscape. However, the paper is a bit long while use specific terminology without explanation, which make me difficult to read it through. I feel this paper should be simpler but use sufficient explanation so that readers who are not familiar with landscape study can also read.

Legends in the maps are small and not clear.

I recommend the authors to discuss more on the validity of the ecological connectivity they calculated for the conservation of the monkey.

It is strange to see Discussion after Conclusion.

Author Response

Point 1:This paper analyzed the changes in landscape spatial pattern between the past and present and tried to evaluate the ecological connectivity of the landscape for the population of a specific monkey. I understand this kind of study is important and the authors can analyze complex geometric patterns of the landscape. However, the paper is a bit long while use specific terminology without explanation, which make me difficult to read it through. I feel this paper should be simpler but use sufficient explanation so that readers who are not familiar with landscape study can also read.

 Response 1: (1) We further summarize some paragraphs in the articles "Analysis of Land Use Characteristics and Ecological Functions in 2.4 Research Area 2000-2020" and "4 Conclusions and Strategies", which are presented in table form (Table 2), which not only simplifies the article, but also makes it easier for readers to understand;(2) We have annotated the specific terms involved in the article, such as landscape matrix, landscape structure, landscape index, landscape pattern, and landscape heterogeneity. Interpretationï¼›(3)The article was revised and professionally edited by MDPI.

Point 2: Legends in the maps are small and not clear.

Response 2: We have revised all the images and added a research roadmap.

Point 3:I recommend the authors to discuss more on the validity of the ecological connectivity they calculated for the conservation of the monkey.

Response 3: In the chapters "3 Analysis of Results" and "4 Conclusions and Strategies", we add content on the ecological significance of ecological connectivity for large land animals such as Yunnan golden snub-nosed monkeys, and the relationship between species pop. 

Point 4:It is strange to see Discussion after Conclusion.

Response 4: First of all. We adjusted the structure of the article, changing "4 conclusions and discussions" to "4 conclusions and strategies" and "5 discussions", and then revised these two parts to add a discussion about Yunnan golden snub-nosed monkeys. Some of the discussions have been deleted and revised.

Reviewer 2 Report

Provide some basic information about the Snub-Nosed Monkey in the introduction (feeding habits, territoriality, predators, competitors, etc.)

Names of rivers in the text and figures must be consistent (e.g. Jinsha versus Yangtze River).

Improve Table 1.

Describe in detail Entropy method, including references.

Methodology could be better understood if you include a flow chart.

Some references related to the information are mentioned but not cited.

Discussion should mention something about ecological aspects of this specie (feeding habits, territoriality, predators, competitors, etc.) and their implications on this study.

Quality of the figures needs to be improved (increase font size, for instance)

Author Response

Point 1:Provide some basic information about the Snub-Nosed Monkey in the introduction (feeding habits, territoriality, predators, competitors, etc.)

 Response 1: Through the adjustment and modification of the article, we have added basic information about the importance of Yunnan golden snub-nosed monkey, species habitat, feeding, species dispersal range, etc. in the "1 Introduction Section", so that readers can better understand the Yunnan golden snub-nosed monkey.

Point 2: Names of rivers in the text and figures must be consistent (e.g. Jinsha versus Yangtze River).

Response 2: We checked all the pictures in the article and unified the names of the rivers.

Point 3:Improve Table 1.

Response 3: We have further refined the entropy method and added technical terminology explanations.

Point 4:Describe in detail Entropy method, including references.

Response 4: We added steps related to the entropy method and supplemented the references.

Point 5:Methodology could be better understood if you include a flow chart.

Response 5:We added "Research Flow Chart" to the research idea in Paper 2.2.

Point 6: Some references related to the information are mentioned but not cited.

Response 6:We adjusted the references and reconfirmed them.

Point 7:Discussion should mention something about ecological aspects of this specie (feeding habits, territoriality, predators, competitors, etc.) and their implications on this study.

 Response 7:Through the adjustment and modification of the article, we added the correlation between the habitat structure characteristics, species habitat, breeding law, species dispersal range, potential predators, interference factors, etc. of Yunnan golden snub-nosed monkey and landscape connectivity in "5 discussions".

Point 8: Quality of the figures needs to be improved (increase font size, for instance)

Response 8: We adjusted and proofread the pixels, legends, and names of all images in the article.

Round 2

Reviewer 1 Report

The paper was revised after the comments by reviewers. Caption in figures are still unclear, may be a problem of my PC. I still feel strange what this Conclusion is for. For me, Conclusion looks better if it is placed at the end.

Author Response

Point 1 :Are the methods adequately described?

Response 1: In order to fully express the logical relationship of each step of the research method of this article, we further sort out the research route of the article to make the research path of this paper clearer.

Point 2: Caption in figures are still unclear.

Response2: We adjusted the image quality to a resolution of 500dpi without affecting the print quality.

Point 3: Is the research design appropriate?

Response 3: The method used in this paper is summarized based on a large number of literature research, and the ecological evaluation and analysis is carried out according to the unique natural environment of the "three parallel rivers" area and the biological characteristics of Yunnan golden snub-nosed monkey, so this method is applicable to this paper.

Point 4: The paper was revised after the comments by reviewers. Caption in figures are still unclear, may be a problem of my PC. I still feel strange what this Conclusion is for. For me, Conclusion looks better if it is placed at the end.

 Response 4:In response to this problem, we have adjusted the structure of the article, swapped "discussion" and "conclusion", and corrected the concluding part of the article to make the concluding part more logical.